# Multi-Physics Coupling Simulation Technique for Phase Stable Cables

Gang Zhang [1], Xiao Chen [1,*], Dazhi Yang [1], Lixin Wang [2], Xin He [1] and Zhehao Zhang [1]

[1] School of Electrical Engineering and Automation, Harbin Institute of Technology, Harbin 150001, China
[2] School of Mechanical Engineering and Automation, Harbin Institute of Technology, Shenzhen 518055, China
* Correspondence: chenxiaoshawn@foxmail.com

**Abstract:** This paper is concerned with calculating the phase stability of a phase stable cable with multi-physics coupling simulations. A three-dimensional electromagnetic-thermal-flow-mechanics multi-physics coupling model is established to simulate the phase stable cable operating in air. The simulation results demonstrate that the electric field distribution of the corrugated cable shows differences from that of a normal coaxial cable around the outer conductor, i.e., the electric field is stronger around the sharp points of the outer conductor than the blunt points, which elevates the voltage endurance requirement of the cable and the risk of breakdown during the transmission of high power. Regarding the thermal deformation, simulations reveal that the corrugated outer conductor restrains the insulation from expanding along both the cross-section direction and the axial direction, and the thermal expansion might be manipulated through the proper design of the outer conductor parameters. Finally, the consistency between the simulated and measured results of phase stability witnesses the validity of the multi-physics coupling model, and its accuracy might be refined with more precise material properties.

**Keywords:** phase stable cable; thermal phase stability; multi-physics coupling simulation; COMSOL





## 1. Introduction

Phase stable cables have been widely used in various phased equipment, such as phased array radars in aircraft or phased array antennas, as well as the feeder line of vector network analyzers [1]. A phase stable cable is adopted to maintain the consistency of the phase difference between its two terminals or the electrical length of the cable, that is, the ratio of its mechanical length to the wavelength of the electromagnetic wave transmitted along it. However, the electrical length of a phase stable cable is not constant but varies slightly due to the effects of diverse environmental stresses, among which mechanical bend and operating temperature are regarded as the two leading causes of phase instability. Generally speaking, mechanical bend only occurs in specific positions, and its effects on electrical length are limited to that local section. On the contrary, the operating temperature of a phase stable cable is usually uniform along its length as a result of the balance between the ambient temperature and the dissipation power of the cable. Therefore, the effects of operating temperature on electrical length take over the cable such that the operating temperature severely influences its phase stability. A resolution to this is to select design parameters and the insulation material precisely, such that the effects of the thermal expansion on electrical length compensate for those of the permittivity variation. For instance, a typical phase stable cable is insulated by polytetrafluoroethylene (PTFE), which has a permittivity lessening with the operating temperature. This paper is dedicated to calculating the phase stability at different operating temperatures with multi-physics coupling simulations.

Multi-physics coupling simulations have been widely used to analyze and optimize cables, even though almost the entire body of research focuses on the heat dissipation of

power cables. The electromagnetic field and thermal field of cable harnesses within more electrical aircraft are calculated in a coupling way with the method of moments (MoM) and the significance of heat effect of crosstalk is illustrated in [2]. Electromagnetic-thermal coupled models with the finite element method (FEM) are applied to optimize spatial geometric parameters of underground power cable systems to attain better heat dissipation efficiency and ampacity [3,4]. An electromagnetic-fluid-temperature multi-physics field coupling model for the drum of a mine cable winding truck is established to analyze the thermal distribution and improve the structure of the drum [5]. Electromagnetic-thermal-flow multi-physics coupling simulations are adopted to calculate the temperature field distribution and the ampacity of submarine power cables, in which a flow field is added to investigate the heat dissipation effect of ocean currents [6]. The effect of a void on the insulation of a crosslinked polyethylene (XLPE) cable [7] and that of particles on the heat dissipation of an ethylene propylene rubber (EPR) cable [8] are studied by electromagnetic-thermal analysis.

In this paper, in addition to electromagnetic-thermal-flow coupling fields, the solid mechanics field of a phase stable cable operating in air is additionally taken into account, such that the phase stability can be simulated holistically in a synthetical model established in the *COMSOL Multiphysics* software of version 5.6. With the inclusion of solid mechanics, the thermal deformation and the interaction between the insulation and the conductors can be analyzed accurately, especially in regard to the phase stable cable with a corrugated outer conductor. Moreover, the thermal phase stability, as a comprehensive result of the temperature-dependent permittivity of the insulation and the thermal deformation, can be calculated precisely in the multi-physics coupling simulation.

The rest of this paper is organized as follows: Section 2 presents the physical bases of the coupling fields, including their governing equations and coupling modes. In Section 3, the processes to model a phase stable cable with a corrugated outer conductor are illustrated. The simulation results are then discussed, and the phase stability is compared with that measured in experiments in Section 4. Section 5 concludes this paper.

## 2. Theories of Multiple Physical Fields and Their Couplings

Regardless of empirical equations, the physical fields and their interactions are analyzed through those physical laws acquired by either inductions or deductions, such that they can be comprehended through solving some specific differential equations, known as the governing equations in the finite element method. For clarity, boundary conditions with respect to a specific model are deferred to Section 3.

### 2.1. Electromagnetic Field

The electromagnetic field is central to our analyses. It is the source of thermal and flow fields and, consequently, the solid mechanics field and the essential condition to determine phase stability. In regard to the calculation of phase differences, it is required to solve the passive damped wave equation of the electric field in the frequency domain as [9]:

$$\nabla^2 \vec{E} - \omega\mu(\mathrm{j}\sigma - \omega\varepsilon)\vec{E} = 0 \tag{1}$$

where $\vec{E}$ is the electric field intensity vector, in which the symbol of a phasor is omitted for convenience, as is applied to other phasors in this paper; $\omega$ is the angular frequency of transmitted electromagnetic waves; $\mu$, $\sigma$, and $\varepsilon$ are, respectively, the permeability, conductivity, and permittivity of a linear isotropic material.

Other field vectors can be calculated from electric field intensity according to Faraday's law of electromagnetic induction in the frequency domain and constitutive equations as follows:

$$\vec{B} = \frac{\mathrm{j}}{\omega}\nabla \times \vec{E} \tag{2}$$

$$\vec{H} = \frac{\vec{B}}{\mu} \tag{3}$$

$$\vec{J}_c = \sigma \vec{E} \tag{4}$$

$$\vec{D} = \varepsilon \vec{E} \tag{5}$$

where $\vec{B}$ is the magnetic flux density vector; $\vec{H}$ is the magnetic field intensity; $\vec{J}_c$ is the conduction current density vector; $\vec{D}$ is the electric displacement vector.

Then, energy density and dissipation power density are accessible through the following equations:

$$w_{\text{eav}} = \frac{\vec{E} \cdot \vec{D}^*}{4} \tag{6}$$

$$w_{\text{mav}} = \frac{\vec{B} \cdot \vec{H}^*}{4} \tag{7}$$

$$q_{\text{eav}} = \frac{\vec{J}_c \cdot \vec{E}^*}{2} \tag{8}$$

where $w_{\text{eav}}$ and $w_{\text{mav}}$ are, respectively, the average energy density of electric and magnetic fields; $q_{\text{eav}}$ is the average dissipation power density caused by electric field in a cycle; $\vec{D}^*$, $\vec{H}^*$, and $\vec{E}^*$ are, respectively, the conjugates of the electric displacement vector, magnetic field intensity vector, and electric field intensity vector. It is noted that the dissipation power density caused by magnetic field, known as the magnetic hysteresis loss, is neglected due to the nonferromagnetic material of interest.

In respect to conductor material, $\sigma$ refers to its actual conductivity, in which Joule heat is the dominant dissipation mode. Nevertheless, as to an insulating material, $\sigma$ is the equivalent joint conductivity of the leaking conductivity and dielectric loss, which can be expressed as follows:

$$\sigma = \sigma_1 + \omega \varepsilon \tan \delta \tag{9}$$

where $\sigma_1$ is the leaking conductivity of the material; $\delta$ is the loss angle of the material; and $\tan \delta$ is the loss angle tangent, which is also named the dissipation factor.

Equation (8) indicates that electromagnetic field affects the thermal field by acting as a heat source. In reverse, thermal field reacts to electromagnetic field by temperature characteristics of physical parameters, including $\sigma$ and $\varepsilon$.

The conductivity of a conductor material can be depicted by a linear resistibility with temperature as follows:

$$\sigma(T) = \frac{\sigma_0}{1 + \alpha_c(T - T_0)} \tag{10}$$

where $T$ is the temperature and $\sigma(T)$ denotes the conductivity at $T$; $T_0$ is the reference temperature, which is often selected as 20 °C or 25 °C; $\sigma_0$ is the conductivity at $T_0$; $\alpha_c$ is called the linear temperature coefficient of resistibility, which is usually regarded as being independent with the temperature. Anyhow, the permittivity of a conductor material is always taken as a constant, which is equal to the permittivity of vacuum.

The effects of the temperature on electromagnetic field via the leaking conductivity of an insulating material can be ignored due to the fact that the leaking conductivity at high frequencies is too low compared to the second item on the right side of Equation (9). However, the permittivity and loss angle of an insulating material are functions of the temperature and, in fact, the frequency, which tightly relate to the material structures. The frequency spectra of the permittivity at different temperatures can be measured in experiments.

### 2.2. Thermal Flow Field

In a solid material, heat dissipates through heat conduction, which can be depicted by Fourier's law as follows [10]:

$$\rho C_{\mathrm{p}} \frac{\partial T}{\partial t} = \nabla \cdot (\lambda \nabla T) + q_{\mathrm{e}} \tag{11}$$

where $\rho$ is the density of the material; $C_{\mathrm{p}}$ is the specific heat capacity at constant pressure; $T$, as aforementioned, is temperature; $t$ is time; $\lambda$ is the heat conductivity; and $q_{\mathrm{e}}$ is the dissipation power of electromagnetic heat. Given that the time constant of the thermal field is much greater than that of the electromagnetic field, $q_{\mathrm{e}}$ is replaced by $q_{\mathrm{eav}}$, the mean of $q_{\mathrm{e}}$ in one cycle.

In a fluid, the thermal field is strongly coupled with the flow field in natural convection, i.e., no flow is driven by any external sources. In this case, heat dissipation results in a rise of the liquid temperature around the solid, leading to a density variation among surrounding fluid. The buoyancy caused by the density variation generates a flow field, which in turn promotes heat dissipation by convection. Hence, natural convection belongs to non-isothermal flow.

The thermal field obeys the heat diffusion equation in a fluid, as follows [10]:

$$\rho C_{\mathrm{p}} \frac{\partial T}{\partial t} + \rho C_{\mathrm{p}} \vec{u} \cdot \nabla T = \nabla \cdot (\lambda \nabla T) \tag{12}$$

where $\vec{u}$ is the flow velocity vector, which manifests the strong dependence of the thermal field on the flow field. Moreover, the flow fluid, represented by flow velocity and pressure, can be depicted as follows [11]:

$$\nabla \cdot \rho \vec{u} + \frac{\partial \rho}{\partial t} = 0 \tag{13}$$

$$\rho \frac{\partial \vec{u}}{\partial t} + \rho \left( \vec{u} \cdot \nabla \right) \vec{u} + \nabla p = \rho \vec{g} + \mu_{\mathrm{f}} \nabla^2 \vec{u} \tag{14}$$

where $p$ is the pressure; $\vec{g}$ is the gravity acceleration vector; $\mu_{\mathrm{f}}$ is the kinematic viscosity of the fluid. Equation (13). reveals the conservation of mass, and Equation (14), which is known as the Navier–Stokes equation, demonstrates the conservation of momentum. In fact, $\rho$ is a function of $T$, and for an approximate ideal gas, it can be expressed as follows:

$$\rho = \frac{p_{\mathrm{A}} M}{RT} \tag{15}$$

where $p_{\mathrm{A}}$ is the absolute pressure; $M$ is the average molar mass of the gas; and $R$ is the universal gas constant, which is equal to 8.314 J·K$^{-1}$·mol$^{-1}$.

Additionally, the surface of the solid material also dissipates heat by the thermal radiation, which is quantified by Stefan–Boltzmann law as follows:

$$\varepsilon_{\mathrm{e}} \sigma_{\mathrm{r}} \left( T^4 - T_{\mathrm{amb}}^4 \right) + \vec{n} \cdot \lambda \nabla T = 0 \tag{16}$$

where $\varepsilon_{\mathrm{e}}$ is the surface emissivity; $\sigma_{\mathrm{r}}$ is Stefan-Boltzmann constant, which is equal to $5.67 \times 10^{-8}$ J·m$^{-2}$·s$^{-1}$·K$^{-4}$; $T_{\mathrm{amb}}$ is the ambient temperature; and $\vec{n}$ is the unit normal vector of the surface.

### 2.3. Thermal Mechanics Field

The deformation of a cable caused by temperature variation is attributed to thermal expansion as a lumped consequence of that of the conductors and the insulation. Hence, the solid mechanics field requires calculating with a thermal stress input. Now that the solid deformation responds much more quickly the thermal variation, it is rational to

take the thermal deformation as a steady state problem. In the case of small deformation, the following equations for elastic materials in steady-state solid mechanics hold [12,13] as follows:

$$\nabla \cdot \vec{\vec{\sigma}} + \vec{f} = \vec{0} \tag{17}$$

$$\vec{\vec{\varepsilon}} = \frac{1}{2}\left(\nabla\vec{d} + \vec{d}\nabla\right) \tag{18}$$

$$\vec{\vec{\varepsilon}} = \frac{1}{2G}\vec{\vec{\sigma}} - \left(\frac{v}{E}\varTheta - \alpha(T - T_0)\right)\vec{\vec{I}} \tag{19}$$

where $\vec{\vec{\sigma}}$ is the stress tensor; $\vec{f}$ is the volume force vector; $\vec{\vec{\varepsilon}}$ is the strain tensor; $\vec{d}$ is the displacement vector; $G$ is the shear modulus; $v$ is Poisson's ratio; $E$ is the Young's modulus; $\varTheta$ is the bulk strain; $\alpha$ is the thermal expansion coefficient, which is regarded as an invariant against temperature; and $\vec{\vec{I}}$ is the identity tensor.

It is noted that the deformation of a cable resulted from thermal expansion actually leads to a distributional variation of the thermal field, of which the effect is neglected due to the fact that the reaction is too modest.

## 3. Modelling of Corrugated Phase Stable Cable

This section may be divided by subheadings. It should provide a concise and precise description of the experimental results, their interpretation, as well as the experimental conclusions that can be drawn.

### 3.1. Geometric Model

In this paper, a phase stable cable with a corrugated outer conductor is of concern, as shown in Figure 1. The cable does not consist of a polymer sheath to the advantage of the heat dissipation, which requires a sealed tube outer conductor rather than a braided one to preserve the insulation. Furthermore, the tube is often corrugated to enhance the flexibility, which leads to the periodic cross-section. It is noted that the cross-sections at different lengths are identical, but show different directions. The cross-section of Figure 1 along the length is illustrated in Figure 2, of which the notations are explained in Table 1.

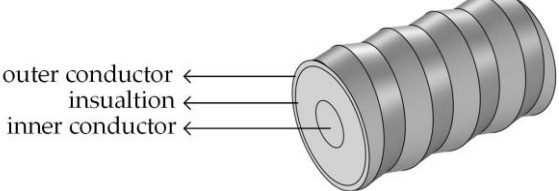

**Figure 1.** Oblique diagram of a section of the phase stable cable with a corrugated outer conductor.

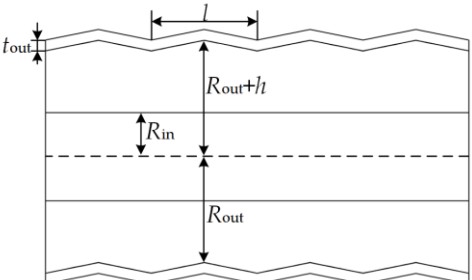

**Figure 2.** Cross-section view of the corrugated cable in Figure 1.

**Table 1.** Explanation of the notations in Figure 2.

| Notation | Meaning | Value |
|:---:|:---:|:---:|
| $R_{in}$ | Radius of the inner conductor | 1.28 mm |
| $R_{out}$ | Minimum inside radius of the outer conductor | 3.32 mm |
| $h$ | Maximum difference of the outer conductor inside radius | 0.3 mm |
| $l$ | Pitch of the corrugation | 3 mm |
| $t_{out}$ | Thickness of the outer conductor along the radial direction | 0.3 mm |

Other than the inner conductor, the periodic geometry of the outer conductor is much more sophisticated if rounded-off details are taken into account. However, it can be represented by less but not smoothly intersectant surfaces without harming the accuracy of later simulations. In this way, if the $x$ axis of a cartesian coordinate system is set along the axis of the inner conductor and the origin is set in the left-end plane in Figure 1, the internal surface of the outer conductor can be depicted by the following parametric equations:

$$\begin{cases} x_1(s_1, s_2) = ls_1 + 0.5ls_2 \\ y_1(s_1, s_2) = R_{out} \sin(2\pi s_1)\left(1 + \frac{hs_2}{R_{out}}\right) \\ z_1(s_1, s_2) = R_{out} \cos(2\pi s_1)\left(1 + \frac{hs_2}{R_{out}}\right) \end{cases}, 0 < s_2 < 1 \qquad (20)$$

$$\begin{cases} x_2(s_1, s_2) = ls_1 - 0.5ls_2 \\ y_2(s_1, s_2) = R_{out} \sin(2\pi s_1)\left(1 + \frac{hs_2}{R_{out}}\right) \\ z_2(s_1, s_2) = R_{out} \cos(2\pi s_1)\left(1 + \frac{hs_2}{R_{out}}\right) \end{cases}, 0 < s_2 < 1 \qquad (21)$$

where $s_1$ and $s_2$ are parameters; and $(x_1, y_1, z_1)$ and $(x_2, y_2, z_2)$ are the coordinates of the points, respectively, in two parts of the surface, which have normal vectors in different directions. The external surface of the outer conductor can be described similarly.

*3.2. Physical Model*

The following assumptions are made in the physical description of the model as follows:

1.  As the skin depth is far less than radii and thicknesses of the conductors at frequencies in interest, the Joule heat dissipated from the conductors can be regarded to yield only on the laminas proximate to the insulation;
2.  The contact thermal resistance between the insulation and the conductors is ignored and the radiation heat transfer of the outer surface of the cable is ignored;
3.  The surrounding air has a temperature of 20 °C and the temperature of the upstream air of natural convection keeps 20 °C;
4.  The velocity of the surrounding air is limited and the flow mode is laminar flow;
5.  The surrounding air is regarded as viscous incompressible fluid, with a Mach number lower than 0.3;
6.  The deformation of the cable satisfies the infinitesimal deformation hypothesis such that linear solid mechanics always holds;
7.  The gravity is ignored when studying the thermal deformation of the cable in solid mechanics.

The conductors and the insulation of the cable are composed, respectively, of copper and expanded PTFE (ePTFE). The material properties of them and surrounding air to be utilized in simulations are listed in Table 2. The relative permittivity of ePTFE dependent on the temperature, $\varepsilon_{rP}(T)$, is measured according to IEC 61196-125 and presented in Figure 3, which is to be utilized in the simulation model with interpolation.

**Table 2.** Material Properties.

| Material | Copper | ePTFE | Air |
|---|---|---|---|
| Relative permittivity | 1 | $\varepsilon_{rP}(T)$ | — |
| Relative permeability | 1 | 1 | — |
| Conductivity at 25 °C | $6 \times 10^7$ | — | — |
| Temperature coefficient of resistibility ($K^{-1}$) | $3.8 \times 10^{-3}$ | — | — |
| Dissipation factor | — | $1 \times 10^{-4}$ | — |
| Density at 20 °C ($kg \cdot m^{-3}$) | 8960 | 2180 | 1.24 |
| Average molar mass ($g \cdot mol^{-1}$) | — | — | 29 |
| Thermal conductivity ($W \cdot m^{-1} \cdot K^{-1}$) | 400 | 0.24 | 0.027 |
| Specific heat capacity ($J \cdot kg^{-1} \cdot K^{-1}$) | 385 | 1000 | 1010 |
| Thermal expansion coefficient ($K^{-1}$) | $1.7 \times 10^{-5}$ | $1 \times 10^{-4}$ | — |
| Young's modulus (Pa) | $1.1 \times 10^{11}$ | $4 \times 10^8$ | — |
| Poisson's ratio | 0.35 | 0.46 | — |
| Kinematic viscosity (Pa·s) | — | — | $4 \times 10^{-5}$ |

Note: A property denoted by "—" means that the property does not exist or not matter in simulations.

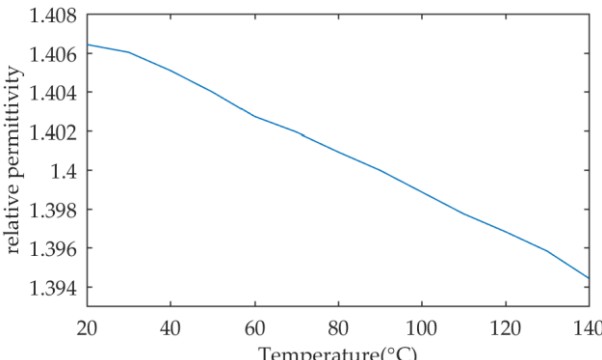

**Figure 3.** Temperature-dependent curve of the relative permittivity of ePTFE at 1 GHz.

### 3.3. Simulation Model

In the multi-physics coupling simulation model, four physical fields, namely electromagnetic field, thermal field, flow field, and solid mechanics field, are considered. A 12 mm long corrugated cable is modeled and the surrounding air represented by a 29 mm × 12 mm × 62 mm air box, the boundaries of which are configured to simulate open space.

In COMSOL, the electromagnetic field is analyzed by the *electromagnetic waves, frequency domain* interface. Only the electromagnetic field of the insulation are solved by FEM, while the conductors are approximated by the impedance boundary condition. The thermal field consists of solid heat transfer and fluid heat transfer. The solid heat transfer only concerns the heat conduction in the cable, and fluid heat transfer deals with heat convection in surrounding air. The flow field in air is calculated by the *laminar flow* interface. Some outside surfaces of the air box are set as open boundary conditions to approximate open space in the fluid heat transfer field and the flow field. In regard to the solid mechanics field, the *solid mechanics* interface is adopted to calculate the deformation of the cable, in which contact pairs are utilized to analyze contact and slide between the insulation and the conductors.

To calculate the phase stability of the cable, three simulation steps are implemented as follows:

1.  Calculate the electromagnetic field, the thermal field and, the flow field when transmitting electromagnetic wave of certain power by a *frequency stationary* study;
2.  Calculate the thermal deformation (solid mechanics field) of the corrugated cable by a *stationary* study;

3.  Calculate the electromagnetic field once again to attain the phase stability of the cable by a *frequency domain* study.

## 4. Analysis of Simulation Results

### 4.1. The Physical Fields

The distribution of electric field in the insulation is shown in Figure 4, in which a 200-W electromagnetic wave is transmitted at 1 GHz. The electric field distribution of the corrugated cable presents notable differences from that of normal coaxial cable, in that the electric field on the outside surface of the insulation does not keep uniform owing to the corrugated shape. To be more specific, the electric field is stronger around sharp points than blunt points, which can be explained by that the sharp points have a higher electron density.

With respect to the insulation, the dissipation power is mainly generated by dielectric loss, which is in direct proportion to the square of electric field intensity. Therefore, the center parts of the insulation have a higher dissipation power than the outer parts.

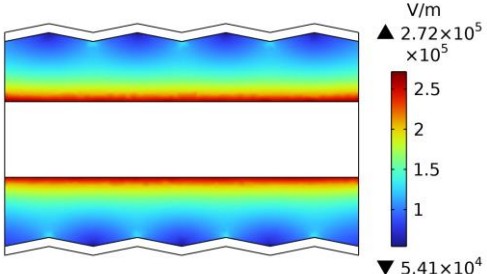

**Figure 4.** Distribution of the electric field in the lengthwise cross-section of the cable insulation.

The distributions of the thermal field and the flow field in the 29 mm × 12 mm × 62 mm air box are illustrated in Figure 5 and produced under a 100 W electromagnetic wave transmitted at 1 GHz. The temperature and flow speed are higher over the cable than in surrounding areas, which indicates that the heat is dissipated to air through natural convection. Compared to the surrounding air, the temperature of the cable is relatively uniform, as shown in Figure 6, which indicates that the heat dispersion of the cable is supposed to enhance by improving convection efficiency of heat convection.

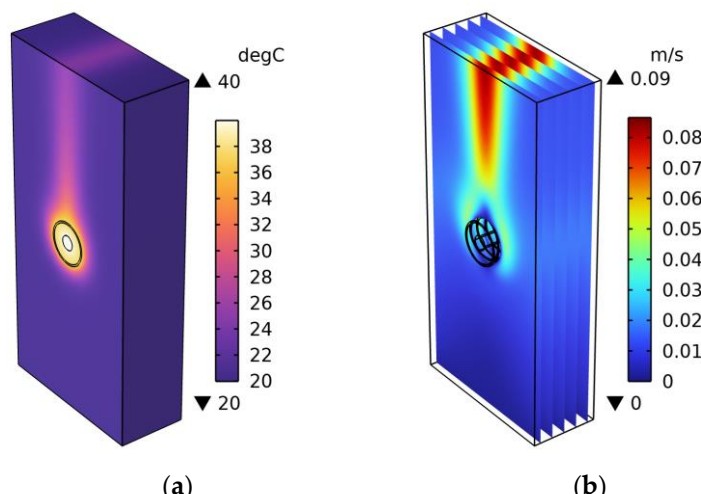

(**a**)                                              (**b**)

**Figure 5.** Distributions of (**a**) the thermal field and (**b**) the flow field when transmitting a 100 W electromagnetic wave at 1 GHz.

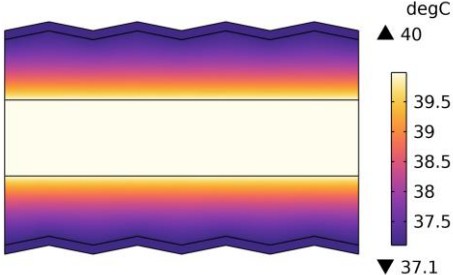

**Figure 6.** Distribution of the temperature of the cable.

The displacement of every particle of the cable is shown as Figure 7, produced under a 100 W electromagnetic wave transmitted at 1 GHz. Due to the difference of the thermal expansion coefficient, the conductors and the insulation have quite different displacements. In fact, the corrugated outer conductor alleviates the expansion of the insulation and also amplifies the stretch of the conductors. Moreover, if the structure parameters of the outer cable are fit such that the expansion of the cable compensates the effect of the temperature variation on phase stability better, the thermal phase stability of the cable would be enhanced.

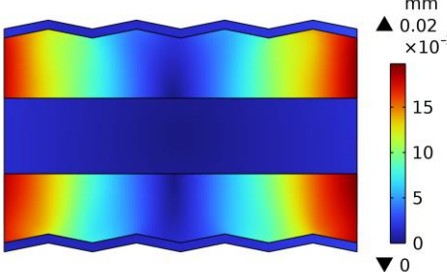

**Figure 7.** Diagram of the displacement of every particle in the lengthwise cross-section of the cable.

*4.2. Phase Stability*

The power of the electromagnetic wave transmitted through the cable at 1 GHz is adjusted to convert the dissipation power and the temperature of the cable. The maximum temperature difference within the cable is less than 3 °C such that the maximum temperature of the cable is regarded as the temperature of the whole cable. The relationship between the transmitted electromagnetic wave power and the temperature of the cable is listed in Table 3.

**Table 3.** The relationship between the electromagnetic wave power and the temperature of the cable.

| EMW Power (W) | Temperature (°C) | EMW Power (W) | Temperature (°C) |
|---|---|---|---|
| 61 | 30 | 519 | 90 |
| 100 | 40 | 602 | 100 |
| 185 | 50 | 684 | 110 |
| 270 | 60 | 752 | 120 |
| 354 | 70 | 827 | 130 |
| 435 | 80 | 897 | 140 |

Meanwhile, the thermal phase stability of a real ePTFE-insulated corrugated phase stable cable, with those parameters listed in Table 1, is determined with vector network analyzer (VNA) in a thermal chamber. As shown in Figure 8, the cable is placed in the thermal chamber and its two terminals are connected to the VNA by feeders. When the temperature in the thermal chamber is altered, the VNA will measure the phase difference between the two terminals at the frequency of interest (i.e., 1 GHz). The phase differences measured at different temperatures are then compared to the phase difference at 20 °C, and the thermal phase stability is extracted and expressed by relative phase changes.

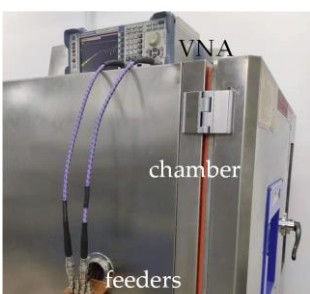

**Figure 8.** Photo of experiment devices used to measure thermal phase stability.

The simulated phase stability results are compared to those obtained in experiments, as shown in Figure 9. It can be seen from Figure 9 that the simulated phase stability agrees with the measured phase stability in general. However, with the rise of the temperature of the cable, the agreement becomes worse, which can be explained by the neglection of the dependence on the temperature of some material properties, such as the thermal expansion coefficient of the insulation. Another group of simulations are implemented regardless of the temperature-dependent permittivity of the insulation, i.e., the permittivity of the insulation at all temperatures is assigned as that at 20 °C, the result of which is also plotted in Figure 9. In comparison with the temperature-independent results, the group of simulations considering the temperature dependency of insulation permittivity have phase stability results that are more aligned to the measured results, which reveals the efficiency of the insulation material on enhancing phase stability.

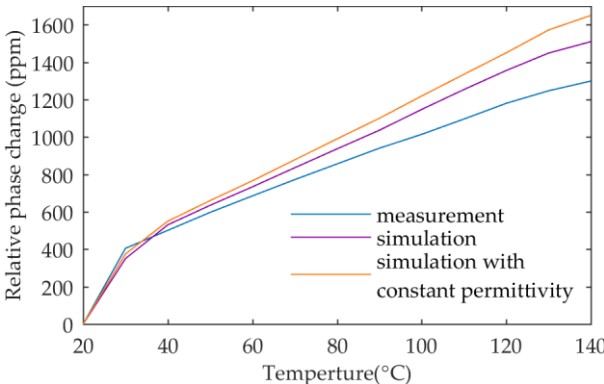

**Figure 9.** Thermal phase stability of the corrugated phase stable cable obtained by multi-physics coupling simulations and experiments.

## 5. Conclusions

In this paper, an electromagnetic-thermal-flow-mechanics multi-physics coupling model for the corrugated phase stable cable is established. These coupling physical fields are solved by COMSOL software using the finite element method. The simulation results indicate the following notes:

The simulated electric field results reveal that the electric field distribution of the corrugated cable is affected by its corrugated outer conductor, which leads to higher electric field intensity in some parts of the corrugated cable than normal cables, and thus increases the voltage endurance as well as the risk of breakdown when transmitting high electromagnetic power.

The displacement or deformation results of the cable evidence the interaction between the conductors and the insulation when the temperature rises. The corrugated outer conductor limits the expansion of the insulation not only in the cross-section plane but also along the axial direction. If the parameters of the conductors and the insulation match well,

the thermal phase stability of the cable is supposed to be enhanced, which illustrates an approach to the refinement of the phase stability.

The phase stability obtained by simulations and that by measurements show good agreements, which demonstrates the validity of the multi-physics coupling simulation applied to the phase stable cable. However, the accuracy of the simulation model may be further improved by utilizing more accurate material and structure parameters.

**Author Contributions:** Methodology, G.Z.; supervision, G.Z. and L.W.; software operation, X.C. and Z.Z.; resource, L.W.; experiment execution, X.H.; data processing, X.H.; manuscript writing, X.C.; manuscript revision, D.Y. All authors have read and agreed to the published version of the manuscript.

**Funding:** This research received no external funding.

**Data Availability Statement:** Some data is contained within the article and other data about the model is available on request due to privacy.

**Conflicts of Interest:** The authors declare no conflict of interest.

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
