# Peer review of "Multi-Physics Coupling Simulation Technique for Phase Stable Cables"

_electronics, doi:10.3390/electronics12071602_

Round 1

Reviewer 1 Report

Dear  Sir

Respect to the paper manuscript entitled “Analysis and Validation of Phase Stable Cables using Multiphysics Coupling SimulationsManuscript ID:    electronics-2267002         , which sent to me to reviewing it for publication in Electronics.  I read the manuscript carefully and patiently. Also, I rebuilt the most part of the calculations and I found that they are correct. The topic of this manuscript can be interest for readers of this valuable Journal. The problem is interesting in this field and the results are important. The manuscript contains a new results in phase stability .

But,  the paper needs  the following points to be noted:                                                                                                             

1-     Punctuations are used randomly. Insert commas or full stops after each and every equation accordingly.

2-     The authors present Equations (1)-(4) without any explanation.

3-      I think the title needs to be reformulated to become more "friendly".

4-     The present form of the abstract is a bit week not much clear. Hence, I recommend to re-write it with 3/4 stronger sentences about your objectives/ findings that will give a better understanding for the readers.

5-     The "Introduction" section should be more concise and some sentences should be rewritten.

6-     What are the innovations or advantages of the proposed model compared to the work of other researchers in the introduction?

7-     Must be corrected some "glitches" of editing.

8-     The conclusion must be improved. 

9-     The figures are drawn even worse. The numbers on the horizontal and vertical coordinates and number axes are too small to be readable. The text size varies between different graphs, and there are even figures that are flattened and distorted. There are also overlaps between the graphs.

10- Suggest the authors cite the related papers in the literature as:

* Two-dimensional problem of generalized magneto-thermoelasticity with temperature dependent elastic moduli for different theories, Multidiscipline Modeling in Materials and Structures, 2009, 5(3), pp. 235–242 https://doi.org/10.1163/157361109789016961

* Analytical solutions of time-fractional heat order for a magneto-photothermal semiconductor medium with Thomson effects and initial stress, Results in Physics, 2020, 18, 103174 https://doi.org/10.1016/j.rinp.2020.103174 

I accept the paper after a minor revision.                                                                                         

Author Response

Thank you for your careful review. The manuscript has been revised according to your comments. Please check our detailed response in the attachment.

Reviewer 2 Report

The manuscript calculates the phase stability of a phase stable cable with multi-physics coupling simulations. It is of interest that electromagnetic heat, natural convection, and thermal expansion, together with temperature-dependent material properties, are utilized to depict the coupling between various physical fields. The motivation is pretty similar to the one recently used nonlocal integrable couplings involving multi-places in soliton theory.  The manuscript is carefully written and the results are clearly described. I recommend publishing it in the journal.  

Author Response

Thank you for your careful review and kind acceptance.

Reviewer 3 Report

the manuscript describes the cable simulation by electromagnetic, heat, and fluid behavior, analyzed by COMSOL. The efficiency of the cable is well mentioned in the manuscript. The followings are comments,

1) behavior of fluid effect the performance of temp and magnetic field. in figure 5, the size of simulation is unknown.

2) radiation effect is large?, by 16)

3) sometimes, the cable is broken by some force. can you explain the cleavage of cable by mechanical simulation?

Author Response

(The authors gave the same response as above.)
